# Sensitivity of Staphylococcal Biofilm to Selected Compounds of Plant Origin

**DOI:** 10.3390/antibiotics10050607

**Published:** 2021-05-20

**Authors:** Denis Swolana, Małgorzata Kępa, Agata Kabała-Dzik, Radosław Dzik, Robert D. Wojtyczka

**Affiliations:** 1Department of Microbiology and Virology, Faculty of Pharmaceutical Sciences in Sosnowiec, Medical University of Silesia in Katowice, ul. Jagiellońska 4, 41-200 Sosnowiec, Poland; dswolana@sum.edu.pl (D.S.); mkepa@sum.edu.pl (M.K.); 2Department of Pathology, Faculty of Pharmaceutical Sciences in Sosnowiec, Medical University of Silesia in Katowice, ul. Ostrogórska 30, 41-200 Sosnowiec, Poland; adzik@sum.edu.pl; 3Department of Biosensors and Processing of Biomedical Signals, Faculty of Biomedical Engineering, Silesian University of Technology, ul. Roosevelta 40, 41-800 Zabrze, Poland; radoslaw.dzik@gmail.com

**Keywords:** antimicrobial activity, anti-biofilm, plants, *Staphylococcus epidermidis*, *Staphylococcus aureus*

## Abstract

*Staphylococcus epidermidis* is a bacterium that belongs to the human microbiota. It is most plentiful on the skin, in the respiratory system, and in the human digestive tract. Moreover, it is the most frequently isolated microorganism belonging to the group of Coagulase Negative Staphylococci (CoNS). In recent years, it has been recognized as an important etiological factor of mainly nosocomial infections and infections related to the cardiovascular system. On the other hand, *Staphylococcus aureus*, responsible for in-hospital and out-of-hospital infections, is posing an increasing problem for clinicians due to its growing resistance to antibiotics. Biofilm produced by both of these staphylococcal species in the course of infection significantly impedes therapy. The ability to produce biofilm hinders the activity of chemotherapeutic agents—the only currently available antimicrobial therapy. This also causes the observed significant increase in bacterial resistance. For this reason, we are constantly looking for new substances that can neutralize microbial cells. In the present review, 58 substances of plant origin with antimicrobial activity against staphylococcal biofilm were replaced. Variable antimicrobial efficacy of the substances was demonstrated, depending on the age of the biofilm. An increase in the activity of the compounds occurred in proportion to increasing their concentration. Appropriate use of the potential of plant-derived compounds as an alternative to antibiotics may represent an important direction of change in the support of antimicrobial therapy.

## 1. Introduction

A biofilm is a population of bacteria that grows on a specific surface (biotic—e.g., tissue or abiotic—e.g., catheter). Bacterial cells, which make up only 10 to 15% of the volume of this structure, are surrounded by the extracellular matrix, making up the remaining 85–90% and consisting of sugars, proteins and extracellular DNA (eDNA) [1,2]. Compared to bacteria living in the planktonic form, microorganisms with the ability to produce biofilm can be up to 1000 times more resistant to the action of antibiotics and antimicrobial agents [3].

Eighty percent of the nosocomial infections are related to bacteria living in biofilm structures. Despite the multiplication of medical procedures to prevent infection, more and more bacterial strains are gaining resistance to the next groups of antibiotics. *Pseudomonas aeruginosa*, which develops a highly resistant biological layer in the respiratory tract of patients with cystic fibrosis, is among the best-known species of bacteria that produce biofilm and are responsible for infections within the human body [4]. Other examples may be bacteria of the *Staphylococcus* genus, including *Staphylococcus epidermidis*, which is statistically the most common cause of infective endocarditis and nosocomial sepsis [5]. Moreover, a *Staphylococcus aureus* resistant to methicillin (MRSA) is responsible for biofilm infections that are more difficult to treat and requires more intensive care [4].

Infections with the above-mentioned microorganisms have been observed in cases of heart valves, orthopedic implants, intravascular catheters, artificial heart, pacemakers, vascular prostheses, cerebrospinal fluid fistulas, urinary catheters, eye prostheses and contact lenses, and intrauterine contraceptive devices [6]. In most cases, they occur as a result of contamination of the biomaterial at the time of implantation or as a result of transient bacteremia [7]. From a medical point of view, both commensal and pathogenic microorganisms form biofilm-like conglomerates that are bound to the epithelial or endothelial lining, embedded in a layer of pulmonary, intestinal or vaginal mucus, attached to teeth or the surface of a medical implant, or formed intracellularly [6]. The surface of the human body, especially the skin, has a microbiota dominated by *S. epidermidis*, which can cause infection, as an opportunistic strain [8].

Transmission may also occur in the course of opening the patient’s gastrointestinal tract or respiratory system. This may be because implants are often used during such surgeries [9]. Bacteria can infect implants quickly upon contact and form a biofilm on their surface, with serious implications for patients. Biofilm-producing microorganisms are much less susceptible to antibiotics and the host’s immune system compared to the planktonic-growing bacteria. Treating such infections is difficult and often poses a challenge to clinicians in the hospital. It also leads to serious complications, i.e., chronic or recurrent infections [6].

Over the past 80 years, many antibiotics have been introduced to the market and have had a positive impact on our lives. For example, in the U.S. alone, nearly 3300 metric tons of antibiotics were sold, of which nearly 1500 tons were penicillins and 500 tons were sulfa drugs, in 2011 alone. The use of antimicrobials in animals may contribute to the emergence of bacterial resistance. That can be transferred to humans, reducing the effectiveness of antimicrobial drugs. Already, AMR (antimicrobial resistance) is recognized as a growing global threat [10].

Treatment of biofilm-associated infections requires relatively high doses and a long-term antimicrobial drug regimen. For this reason, there is a non-zero risk of developing antimicrobial resistance (AMR). Because of these, it becomes imperative to search for more effective biofilm inhibiting agents [11].

## 2. Staphylococcal Biofilm

*Staphylococcus aureus* and coagulase-negative staphylococci are considered the most common cause of biofilm-forming infections [12]. Approximately 20–25% of the human population carries *Staphylococcus aureus*, and it has been shown that there is a strong causal link between nasal carriage and an increased risk of nosocomial infection in these individuals. Due to the increasing share of this pathogen in infections, their rapid development, and the possibility of transforming into a chronic, persistent and recurrent infection, this microorganism deserves special attention [13].

In addition to many pathogenic factors of cutaneous staphylococcus, such as the ability to produce toxins, excluding the increased resistance of these bacteria to the antibiotic treatment used, as well as the possibility of secreting immune-evasion proteins, difficulties in treatment and the increasing incidence of chronic infections are conditioned by its ability to produce biofilm [14,15,16]. Most authors describe the process of biofilm formation as three main stages: adhesion, maturation, and dispersion [14]. The stages of biofilm formation are shown in Figure 1.

Although biofilm can be formed from one cell, environmental conditions, such as the diversified supply of oxygen, nutrients, or electron acceptors determine the diversity of the cell population [13].

## 3. Plant-Derived Antibiofilm Substances

The development of new antibiotics allows their use in the course of patient treatment. However, the discovery of antibiotics is becoming increasingly difficult. It is therefore necessary to search for unconventional treatment strategies as well as alternatives to these common antimicrobial agents. In addition to bacteriophage therapy, photodynamic treatment and numerous derivatives of chemical compounds, raw materials of natural origin remain an important source of therapeutic substances. Plants and microorganisms produce a wide range of diverse secondary metabolites that serve as defenses against pathogens. This is the source of many substances against a variety of bacterial virulence factors [17].

Already, more than 80% of medicinal substances are directly derived from natural substances or have been produced from natural products. Available data indicate that about 50% of pharmaceuticals contain active substances synthesized from previously identified or isolated compounds derived from plants or animals [18].

Active agents found in plants can be divided into two main groups. The first includes products of primary metabolism, including carbohydrates (sugars, mucilages), fats (fatty acids, phytosterols), proteins, amino acids, vitamins, enzymes, and pigments. The second group consists of secondary metabolism products, including glycosides, terpenes, saponins, polyphenols, alkaloids, essential oils, organic acids, and others [19]. Antimicrobial activity is mainly exhibited by secondary metabolites present in substances of plant origin. They have a wide range of action, depending on the species of plant from which they originate or the climate of the country where it occurs [20]. Depending on their structure, they exhibit various biological properties ranging from antioxidant, antibacterial, antifungal to modulating enzymatic activity. Considering the above plants are excellent sources of novel antimicrobial compounds [21].

Phenols and polyphenols, as the simplest phytochemicals, have a strong ability to bind various macromolecules such as proteins or glycoproteins and in this form are toxic to microorganisms. They can also enhance the effects of antibiotics, especially against Gram-positive bacteria [22]. Quinones are ubiquitous and highly reactive organic compounds. The free radicals they produce, form irreversible complexes with microbial proteins. In turn, anthraquinones inactivate bacterial adhesins and polypeptides, leading to bacterial cell dysfunction [20]. Flavonoids are a structurally diverse group of compounds produced by plants, among others, in response to bacterial infection. Their chemical structure differentiates them into: flavones, flavonols, flavanones, flavanonols, anthocyanides, isoflavones, and chalcones. The activity of these compounds is due to their ability to form complexes with proteins and bacterial membranes, induce oxidative stress, or inhibit electron transport in the bacterial respiratory chain [20,23]. Tannins are polymeric phenolic substances that exhibit antimicrobial activity through inactivation of enzymes, adhesins, and transport proteins, and antiperoxidative properties [20]. Terpenes and terpenoids are in turn produced by plants to interact with other organisms. Their high concentrations are also found in essential oils. In addition to the aforementioned characteristics, antimicrobial and antibiofilm activities of these substances have also been demonstrated, either by the mechanism of additive or synergistic action with other antimicrobial drugs [20,24]. The proven antimicrobial activity of coumarins is mainly due to their ability to bind to the beta subunit of DNA gyrase and block the ATPase activity of bacteria. They are composed of a combined benzene and pyrone ring [20,25]. Alkaloids are compounds containing a nitrogen atom in their structure. Their antimicrobial activity is based on disruption of growth, bacterial proliferation, accumulation in the bacterial cell, and DNA intercalation [20]. The diverse mechanism of action and numerous in vitro studies for the above-mentioned groups of compounds open up a wide range of possibilities for the use of these antimicrobial agents in therapy, given the ever-increasing resistance to commonly used antibiotics [22].

Table 1 presents the plant natural compounds discussed in the literature that may find application in antimicrobial and antibiofilm therapy. The information was collected from scientific papers published between 2010 and 2020, presenting the use of plant natural compounds in counteracting biofilm formation. Substances from organisms included in other systematic groups were excluded.

Analyzing the presented results of published studies, it can be observed that the concentrations of active substances contained in products of plant origin are within the range of 0.1 to several thousand µg/mL, which indicates a large variation in their activity against biofilm. Different antimicrobial efficacy of the analyzed substances was also found depending on the age of the biofilm [27,29,30,32,34,36,44,60]. Mature structures are difficult to eliminate due to the accumulation of extracellular matrix and altered cellular metabolic activity. Such a phenomenon was noted, among others, in the study of Gondil et al. [75]. The second important feature of the analyzed substances is the increase of their activity, occurring proportionally to the increase of their concentration. Such a fact was observed for many substances, e.g., for psychorubrin, which caused inhibition of mature biofilms of *S. aureus* ATCC 33591, as well as for aurantioglycoladin, α-mangostin, baicalein, and luteolin [41,42,43,46,48]. Similarly, aurantioglycoladin induced a concentration-dependent inhibition of biofilm production of *S. epidermidis* ATCC 35984. The inhibition of biofilm growth observed in some cases at concentrations of extracts higher than the MIC and even the MBC, indicated that bacterial cells in biofilm are more resistant to antimicrobials, compared to cells growing in planktonic form. This is a well-known feature, confirmed in the present work for cinnamaldehyde, diterpenes, terpenoids, some flavones, norigenin, *Vernonia condensata* leaf extract, *Olea europaea* extract, and rhodomyrton [27,32,36,46,51,53,65,73]. Another observed phenomenon confirming the reduced activity of substances in relation to mature biofilm is the inhibition of this structure formation, occurring at a higher level than its eradication in the same concentration of the active substance—particularly evident in the case of 1-monolaurin, nerolidol, (+)-dehydroabietic acid, thymol, rosemary oil, and cinnamaldehyde [26,27,29,36,60].

Within substances classified systematically into one chemical group, their antibiofilm activity was also significantly differentiated and required the use of individual substances in concentrations differing by several orders of magnitude to achieve the same efficacy. Such a phenomenon was observed for sesquiterpenes and diterpenes occurring in oleoresin from *Copaifera duckei*, methanol extract from aerial parts of *Anthemis stiparum* subsp. *sabulicola* as well as eugenol and nerolidol. The aforementioned substances, belonging to terpenes, showed differential activity against MRSA strains [28,29,30,31]. In the case of chalcones: xanthohumol and desmethylxanthohumol, extracted from the plant *Humulus lupulus*, the effectiveness of their action against mature and immature biofilm was differentiated. The concentrations of these compounds, resulting in inhibition of biofilm production, differed approximately tenfold [44]. A similar phenomenon, but of lower intensity, occurred for andrographolide and (+)-dehydroabietic acid, which both belong to the group of terpenoids. The inhibitory concentration of the formed biofilm for the mentioned compounds differed two-fold [33,36].

Luteolin, applied to a single-species *S. aureus* biofilm and a dual-species biofilm formed by *S. aureus* and *L. monocytogenes*, showed greater antibiofilm activity against the former. Such a phenomenon confirms the greater luteolin resistance of the dual-species biofilm compared to the single-species biofilm [48].

During the analysis, it was noted that flavonostilbenes exhibited stronger antimicrobial activity than flavonoids. The lowest growth inhibitory concentration (MIC) of naringenin (flavonoid) for *S. aureus* was 120-fold higher than the MIC for *S. epidermidis* of allopecurones (flavonostilbenes) [51,57]. Alopecuron D (flavonostilbenes) showed activity against biofilm formation at lower concentrations, but did not lead to death of all microorganisms [57]. On the other hand, diterpenes were more effective against the tested biofilm-forming pathogens compared to flavonoids, as evidenced by a higher percentage of reduction in biofilm formation [32].

The differential activity of plant-derived substances depending on the bacterial species was also confirmed. Differential activity of chelerythrine and sanguinarine was observed against *S. aureus* and *S. epidermidis*—twice lower EC_50_ for chelerythrine and six times lower EC_50_ for sanguinarine in case of *S. epidermidis* [66]. The terpenes contained in oleoresin from *Copaifera duckei* showed differential activity against *S. aureus* and *S. epidermidis*—MBIC about 12 times higher for *S. aureus* [30]. MIC analysis of methanolic extract from aerial parts of *Anthemis stiparum* subsp. *Sabulicola* showed about 25 times higher antibacterial activity against *S. epidermidis*, compared to *S. aureus* [31].

## 4. Conclusions

An important aspect of the analyzed substances of plant origin is their ability to have other biological activity, in addition to the presented antimicrobial activity. These substances possess anti-inflammatory, anticancer, and antioxidant properties [65,76,77,78,79,80,81,82,83]. The use of natural compounds in therapy is promising due to the occurrence of low natural resistance of microorganisms, but it carries some uncertainties. These include failure of therapy due to uncontrolled microbial growth or change in bacterial virulence. Natural antimicrobial compounds can also weaken the microbiota, causing bacterial dysbiosis, which is dangerous for the body. An ideal antimicrobial should selectively reduce the virulence-determining factors of a strain, without any toxicity to the macroorganism or the viability of the bacteria that make up its natural microbiota. The combination of multi-targeted action of antibiotics and antimicrobial substances has shown promising results against biofilm-producing bacteria. Interference of natural substances into host biochemical pathways is also encountered under certain conditions, which induces an increase in bacterial resistance to the host immune system. Structural optimization of natural products requires improved antimicrobial effects and reduced side effects based on knowledge of the mechanism of bacterial virulence, antibiotic resistance, and bacteria–host interactions. Natural antimicrobial products are a promising alternative as enhancers of drugs used against antibiotic-resistant bacteria [17].

## Figures and Tables

**Figure 1 antibiotics-10-00607-f001:**
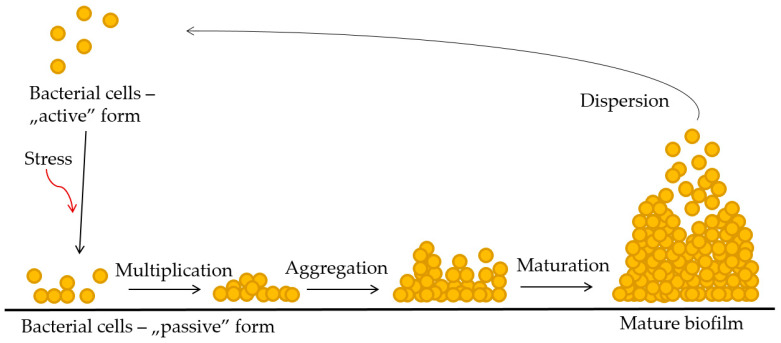
Stages of biofilm formation in staphylococci [4].

**Table 1 antibiotics-10-00607-t001:** Plant-derived antimicrobial substances with antibiofilm activity.

No	Substance	Source	Systematic Group	Scope of Activity	Ref.
1.	1-monolaurin	coconut oil	glycerides	Inhibition of biofilm formation at 500 µg/mL at 96.78%, biofilm eradication at 500 µg/mL among clinical isolates of *S. epidermidis* (the collection of Microbiology Laboratory Faculty of Medicine, Public Health, and Nursing UGM) at 68.16%	[26]
2.	Cinnamaldehyde	essential oil of *Cinnamomum* Scheffer	aldehydes	Inhibition of *S. aureus* ATCC 25923 biofilm formation at a concentration of 0.5 mg/mL with a decrease of 4 logarithmic values and the ability to eliminate mature biofilm approximately 100-fold at a concentration of 0.5 mg/mL	[27]
3.	Eugenol	*Dianthus caryophyllus L.*	terpenes	MIC = 0.04%; reduction by more than 50% of MRSA (Culture Collection of Antimicrobial Resistant Microbes, Seoul, Korea) and MSSA ATCC 29213 biofilm growth in vitro at 1/2 MIC concentration	[28]
4.	Nerolidol	*Pogostemon heyneanus*	terpenes	MIC = 0.025%; growth reduction of immature MRSA (clinical strains) biofilms at 1/2 MIC and 1/4 MIC at 88%, while inhibition of mature MRSA (clinical strains) biofilms at 1/2 MIC and 1/4 MIC at 85%	[29]
5.	Sesquiterpenes and diterpenes	oleoresin of *Copaifera duckei*	terpenes	IC_50_ value for mature *S. aureus* (clinical isolate) biofilms exposed to oleoresin = 21.85 µg/mL; MBIC for *S. epidermidis* (water isolate) = 12.50 µg/mL, MBIC for *S. aureus* (clinical isolate) between 0.78 and 100.00 µg/mL	[30]
6.	Methanol extract from aerial parts of *Anthemis stiparum* subsp. *Sabulicola*	aerial parts of *Anthemis stiparum* subsp. *sabulicola*	terpenes	MIC = 1.56 mg/mL for *S. aureus* subsp. *aureus* ATCC 25923; 59.06% inhibition on biofilm growth at the MIC concentration; MIC = 25 mg/mL for *S. epidermidis* MU 30, 30.43% inhibition on biofilm growth at the MIC concentration	[31]
7.	Organic extract of *Prunus cerasoides*	*Prunus cerasoides*	diterpenes	MIC = 5 mg/mL; inhibition of approximately 50% of mature *S. aureus* MTCC 740 biofilm growth; MIC = 10 mg/mL for *S. epidermidis* MTCC 435; MIC = 1 mg/mL for MRSA (clinical strain)	[32]
8.	Andrographolide	*Andrographis paniculata*	terpenoids	Inhibition of *S. aureus* MTCC 96 biofilm growth by about 45% on the polystyrene surface after 24 h of exposure to the compound at a concentration of 50 μg/mL	[33]
9.	Celastrol	Extract of *Tripterygium wilfordii* and *Celastrus regelii*	terpenoids	Inhibition of biofilm formation by 25.5–85.07%, eradication of mature biofilm by 40.5–80.2% for *S. aureus* (MSSA) ATCC 29213; inhibition of biofilm formation by 27–89.3%, eradication of mature biofilm by 49.5–82.8% for *S. aureus* (MRSA) clinical strains	[34]
10.	Emulsion containing resin acids	Norway spruce—*Picea abies*	terpenoids	90.8% ± 8.4% growth inhibition of *S. aureus* MRSA (ATCC BAA-44); significant increase in eradication and reduction in biofilm formation for *S. aureus* Mu50 and *S. epidermidis* ET013	[35]
11.	(+)-dehydroabietic acid	oleoresin from a tree of the genus *Picea*	terpenoids	MIC = 21 mg/L for *S. aureus* ATCC 25923; significant inhibition of biofilm formation (IC_50_ = 8.35 mg/L) and action on biofilm-forming *S. aureus* (IC_50_ = 33.9 mg/mL)	[36]
12.	Phosprenil	conifer needles of fir (*Abiessibirica*) or pine (*Pinussylvestris*)	prenoles	Approximately 2-fold inhibition of *S. aureus* ATCC 6538 and a clinical strain 010Ng, biofilm growth at concentrations of 7.5–30 mg/mL	[37]
13.	Carvacrol	*Plectranthus amboinicus*	phenols	MIC = 0.25 mg/mL for *S. aureus* OVRSA and ATCC 6538; antibiofilm activity against *S. aureus* ATCC 6538 at 0.25 mg/mL; biofilm reduction ability at all carvacrol concentrations tested (0.062 to 4 mg/mL)	[38]
14.	Carvacrol	oregano oil	phenols	Reduction in *S. aureus* BMA/FR/032/0074 biofilm production at a concentration of 0.50–1.00 mM	[39]
15.	Carvacrol	essential oils of oregano, thyme	phenols	Ability to approximately 1000-fold eliminate mature *S. aureus* ATCC 25923 biofilm at 0.5 mg/mL and inhibit its formation at the same concentration, with a decrease in CFU of 1,000,000/mL	[27]
16.	Thymol	essential oils of *Thymus* and savory	phenols	Ability to approximately 1000-fold eliminate mature *S. aureus* ATCC 25923 biofilm at a concentration of 0.5 mg/mL and inhibit its formation at a concentration of 0.5 mg/mL with a decrease of 5 logarithmic values	[27]
17.	Ellagic acid xyloside	*Rubus ulmifolius*	polyphenols	50% inhibition of *S. aureus* (MSSA) osteomyelitis isolate (UAMS-1) biofilm formation at a concentration of 64 µg/mL	[40]
18.	Ellagic acid rhamnoside	*Rubus ulmifolius*	polyphenols	50% inhibition of *S. aureus* (MSSA) osteomyelitis isolate (UAMS-1) biofilm formation at a concentration of 64 µg/mL; capable of 90% inhibition of biofilm formation at a concentration of 128 µg/mL	[40]
19.	Psychorubrine	*Mitracarpus frigidus*	quinones	Inhibition of mature biofilms in approximately 56% (MIC) and 46% (1/2 MIC) for *S. aureus* (MRSA) ATCC 33591 and in 84% (MIC) and 85% (1/2 MIC) for *S. aureus* (MRSA) ATCC 33592	[41]
20.	Aurantioglycoladine	*Clonostachys candelabrum*	quinones	MIC = 64 µg/mL for *S. epidermidis* ATCC 35984; inhibition of biofilm production in 55% at a concentration of 256 μg/mL, in 51% at 64 μg/mL, in 19% at 32 μg/mL and in 10% at 16 μg/mL; MIC = 300 µg/mL for *Staphylococcus aureus* DSM 1104	[42]
21.	Alpha-mangostin	pericarp of Garcinia mangostana L. (family *Clusiaceae*)	xanthones	Significant inhibition of *S. aureus* (MRSA) standard isolate DMST 20654 biofilm formation in a dose-dependent manner from 1/16 MIC to MIC; at 1/2 MIC, inhibition of biofilm formation by approximately 70%	[43]
22.	Xanthohumol	*Humulus lupulus*	chalcones	Inhibition of *S. aureus* (clinical isolate T28.1) biofilm-forming ability and ability to reduce existing biofilm at a concentration of 39 µg/mL (MIC)	[44]
23.	Desmethylxanthohumol	*Humulus lupulus*	chalcones	Inhibition of the biofilm-producing ability of *S. aureus* (clinical isolate T28.1) at a concentration of 4.9 µg/mL (1/2 MIC) and the ability to destroy an existing biofilm at a concentration of 2.45 µg/mL (1/4 MIC)	[44]
24.	Resveratrol	Peanuts (*Arachis hypogea*), blueberries and cranberries (*Vacciniumspp*.), Japanese knotweed (*Polygonum cuspidatum*), grapevine (*Vitis vinifera*)	stilbenes	MIC = 350 µg/mL; inhibition of *S. aureus* (clinical MRSA isolate) biofilm formation by approximately 39.85% at a concentration of 100 µg/mL	[45]
25.	Baicalin	*Astragalusmembranaceus* root	flavones	Inhibition of *S. aureus* (SA002, isolated from the nose swab of a pig with pneumonia) biofilm formation in a dose-dependent manner, statistically significant reduction in increase in MIC and 5 MIC	[46]
26.	5-hydroxy-3,7,4′-trimethoxyflavone	*Chromolaena odorata* (*Asteraceae*)	flavones	Inhibition of *S. aureus* ATCC 29213 (MSSA,) biofilm production at a concentration of 1 mg/mL, with activity greater than 50% after 24 h	[47]
27.	Luteolin	broccoli, peppers, thyme and celery	flavones	MIC = 16 µg/mL for *S. aureus* ATCC 25923; MIC = 64 µg/mL for two *S. aureus* clinical strains from derived from raw goat milk; concentration-dependent anti-biofilm activity against *S. aureus* ATCC 25923 biofilm at concentrations of 1/8 MIC and above; antibiofilm activity of luteolin against dual-species biofilm of *S. aureus* ATCC 25923 and *L. monocytogenes* ATCC 19115 (MIC 32 µg/mL) at concentrations of 1/4 MIC and above	[48]
28.	Dihydrovogonin	bird cherry extract *Prunus avium*	flavones	Inhibition of growth of planktonic form at concentrations of 125–500 µg/mL; reduction in *S. aureus* (CIP 53.154) biofilm mass correlated with a decrease in the number of bacteria in the forming biofilm in the concentration range of 125–500 µg/mL	[49]
29.	Moryna	figi, migdały	flavones	Inhibition of biofilm formation and elimination of the formed structure for clinical isolated cultures of MRSA (MBIC = 281.83 μg/mL) and VRSA (MBIC = 398.10 μg/mL)	[50]
30.	Organic extract of *Prunus cerasoides*	*Prunus cerasoides*	flavonoids	MIC = 1 mg/mL for *S. aureus* MTCC 740; MIC = 10 mg/mL for *S. epidermidis* MTCC 435; inhibition of mature *S. aureus* MTCC 740 biofilm at 86.5 mg/mL by approximately 45%	[32]
31.	Naringenin	hemp (*Cannabis sativa* L.)	flavonoids	MIC = 512 µg/mL for *S. aureus* clinical strain; minimum biofilm eradication concentration MBEC = 2048 µg/mL	[51]
32.	Derriobtusone A	root bark of *Lonchocarpus obtusus*	flavonoids	Rapid decrease in biomass and CFU of *S. aureus* JKD 6008 biofilm at concentrations of 250 and 500 µg/mL	[52]
33.	Ethyl acetate fraction of *Vernonia condensata* leaf extract	leaves of *Vernonia condensata*	flavonoids	Inhibitory effect of MIC, 2 MIC and 4 MIC concentrations on adhesion of *S. aureus* (MSSA) ATCC 25923 and *S. aureus* (MRSA) ATCC 1485279—inhibition in the range from 60% to 100%	[53]
34.	Corilagin	fruit of *Terminalia chebula* Retz	tannins	Decrease in cell adhesion for *S. aureus* ATCC 11632: IC_50_ = 3.18 μg/mL	[54]
35.	Tannic acid	*Quercus infectoria* G. Olivier extract	tannins	Inhibition of MRSA (NPRC R001-R047, clinical strain) biofilm formation at MIC (0.13–0.50 µg/mL) and sub-MIC concentrations; inhibition of MSSA (NPRC S001-S050, were isolated from nasal specimens of healthy volunteers) biofilm formation at MIC (0.13–0.50 µg/mL)	[55]
36.	Hamamelitanin	whISOBAX, witch hazel extract (*Hamamelis virginiana*)	tannins	Reduction in *S. epidermidis* ATCC 35984 biofilm formation by nearly 50% at a 1:26 dilution	[56]
37.	Alopecuron H, I, J, K, L, A, B, D, soforaflavone G	root of *Sophora alopecuroides*	flavonostilbenes	MIC 6.25–3.125 µg/mL; inhibition of *S. epidermidis* ATCC 35984 biofilm formation; preventing biofilm formation at lower concentrations without bactericidal activity	[57]
38.	Hyperforin in the form of dicyclohexylammonium salt	*Hypericum perforatum*	phloroglucinols	MBIC = 25 µg/mL for *S. aureus* (ATCC 29213; ATCC 43300 and Ig5—clinical isolate); inhibition of biofilm development by 21–45%	[58]
39.	Thyme oil	*Thymus vulgaris*	essential oils	MIC = 0.078% for *S. aureus* ATCC 25923; 71% reduction in *S. aureus* ATCC 25923 biofilm viability at a concentration corresponding to the MIC	[59]
40.	Essential oil	hemp (*Cannabis sativa* L.)	essential oils	MBEC = 24 mg/mL for *S. aureus* (MSSA) ATCC 29213	[51]
41.	Essential oil from the leaves and stem of *Plectranthus amboinicus*	*Plectranthus amboinicus*	essential oils	Antibiofilm activity against *S. aureus* OVRSA and ATCC 6538 at 0.5 mg/mL; inhibition potential against *S. aureus* ATCC 6538 at all essential oil concentrations tested (0.062–4 mg/mL)	[38]
42.	Essential oil	*Rosmarinus officinalis* L.	essential oils	MIC 1.25–2.5 µL/mL for *S. aureus* ATCC 9144; MIC 0.312–0.625 µL/mL for *S. epidermidis* S61; inhibition of *S. epidermidis* S61 biofilm production above 57% at a concentration of 25 μL/mL; biofilm eradication at a concentration of 50 μL/mL	[60]
43.	Essential oil from the aerial parts of *Anthemis stiparum* subsp. *Sabulicola*	aerial parts of *Anthemis stiparum* subsp. *sabulicola*	essential oils	Inhibition of biofilm formation of *S. epidermidis* MU 30 and *S. aureus* ATCC 25923 to 29.17% and 8.25%, respectively, at a concentration of 25 μL/mL	[31]
44.	Essential oils	*Pogostemon heyneanus* and *Cinnamomum tamala*	essential oils	MIC = 2–6%; inhibition of immature biofilms of MRSA (clinical strains) at concentrations of 3–0.5% with efficacy of 55–80%; for biofilms of mature MRSA, inhibition of 60–80%	[29]
45.	Ethanolic leaf extract of *Mangifera indica L.*	leaves of *Mangifera indica L.*	tannins	Reduction of mature biofilm of eight *Staphylococcus* spp. strains from cows with mastitis by ethanol extract at a concentration of 45.3 mg/mL	[61]
46.	Erianin	*Dendrobium chrysotoxum*	natural bibenzyl compound	Significant decrease in *S. aureus* (strain Newman D2C—ATCC 25904) biofilm formation at a concentration of 64 µg/mL	[62]
47.	Chilean tree fruit extract of Arrayan and Peumo	Arrayan [*Luma apiculata* (DC.) Burret.] and Peumo [*Cryptocarya alba* (Molina) Looser]	flavonols, anthocyanins	Higher activity of Arrayan extract (IC_50_ = 0.229 ± 0.017 mg/mL) compared to Peumo extract (IC_50_ = 0.473 ± 0.028) against biofilm of *S. aureus* ATCC 25923	[63]
48.	Polyphenolic extracts from cladodes	*Opuntia ficus-indica* (L.)Mill.	phenolic acids and flavonols	Significant inhibition of *S. aureus* ATCC 35556 biofilm formation by extracts from mature and immature clades at a concentration of 1500 µg/mL	[64]
49.	Extracts of Tunisian varieties of *Olea europaea* L., i.e., „Chetoui”, „Meski”, „Oueslati” and „Jarboui”	*Olea europaea* L.	phenols and flavonoids	Best antibiofilm activity of Chetoui and Meski extracts against *S. aureus* strains (MRSA and *S. aureus* ATCC 25923) with inhibition values >50% at MIC doses and 72–89.8% at doses of 2 MIC; good antibiofilm activity of Jarboui and Oueslati extracts against tested bacterial *S. aureus* strains (MRSA and *S. aureus* ATCC 25923) in the range from 54.5 to 83.8% at the concentration of 2 MIC	[65]
50.	Cheleritrin, sanguinarine	*Krameria lappacea, Aesculus hippocastanum* and *Chelidonium majus*	flavonoids, alkaloids	1.3 to 5.5 times inhibition of mature *S. aureus* ATCC 6538P and *S. epidermidis* ATCC 35984 biofilm formation and eradication; EC_50_ of cheleritrin for *S. aureus* (ATCC 6538P reference strain)—15.2 ± 2.3 µM, for *S. epidermidis* ATCC 35984—8.6 ± 0.4 µM; EC_50_ of sanguinarine for *S. aureus* ATCC 6538P—24.5 ± 3.6 µM, for *S. epidermidis* ATCC 35984—4.4 ± 1.3 µM	[66]
51.	Alcoholic extract	*Cytinus hypocistis* and *Cytinus ruber*	flavanoids, phenols	Inhibition of biofilm formation in 60–80% at 1/2 MIC for *S. epidermidis* ATCC 35984	[67]
52.	Ethanol extract	leaves of *Moringa stenopetala*	esters, alcohols, fatty acids and others	Antibiofilm activity and inhibition of MRSA (three clinical strains isolated from HIV infected patients) biofilm production at a concentration of 1000 μg/mL	[68]
53.	Tanreqing injection	*Scutellariae radix, Lonicerae flos, Forsythiae fructus, Ursi fel, Naemorhedi cornu*	flavonoids, phenols and others	MIC = 4125 μg/mL for MRSA ATCC 43300; strong reduction in bacterial viability in mature MRSA biofilms at 1/2 MIC and 1/4 MIC	[69]
54.	Alcoholic extract	*Zanthoxylum armatum* DC.	alkaloids and others	> 50% inhibition of *S. aureus* UAMS-1 biofilm formation at 256 μg/mL, resulting from overall growth inhibition at this dose (IC_50_ = 32–256 μg/mL)	[70]
55.	Essential oil	*Rhanterium suaveolens*	alcohols, aldehydes and others	Highest antibiofilm activity of 50.3% against *S. epidermidis* MU30 at 20 μg/mL essential oil	[71]
56.	Aqueous plant extracts	branches of *Bauhinia acuruana,* fruits of *Bauhinia acuruana,* leaves of *Pityrocarpa moniliformis,* stem bark of *Commiphora leptophloeos*	polyphenols coumarins, terpenes	Inhibition of biofilm production of *S. epidermidis* ATCC 35984 at a concentration of 4 mg/mL, in a range of approximately 77–85%	[72]
57.	Rhodomyrtone	*Rhodomyrtus tomentosa*	-	MIC = 0.25–1 µg/mL for *S. aureus* and *S. epidermidis* clinical isolates; at 0.5 MIC and 0.25 MIC was found to be effective in reducing biofilm formation in most of the *S. aureus* isolates, At 0.5 MIC rhodomyrtone reduced biofilm formation in all six *S. epidermidis* isolates, bactericidal effect in mature biofilm at 64 MIC for *S. epidermidis*; rhodomyrtone demonstrated better activity in killing the organisms in 24 h biofilms than those in 5-day biofilms	[73]
58.	Skeletocutins A-L	*Skeletocutis* sp. (MUCL56074)	-	Inhibition of *S. aureus* DSM1104 biofilm formation by skeletocutin I: up to 86% at a concentration of 256 μg/mL and up to 28% at a concentration of 150 μg/mL	[74]

MIC (minimal inhibitory concentration); MSSA (methicillin sensitive *Staphylococcus aureus*); MBIC (minimal biofilm inhibitory concentration); IC_50_ (half maximal inhibitory concentration); EC_50_ (half maximal effective concentration); MTR (multidrug-resistance); VRSA (vancomycin-resistant *Staphylococcus aureus*).

## Data Availability

No new data were created or analyzed in this study. Data sharing is not applicable to this article.

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
