# Peer review of "Sensitivity of Staphylococcal Biofilm to Selected Compounds of Plant Origin"

_antibiotics, 2021, doi:10.3390/antibiotics10050607_

Round 1

Reviewer 1 Report

The article by Svolana et al. and entitled “Sensitivity of staphylococcal biofilm to selected compounds of plants origin” is a review article that aims to describe a series of plant-derived compounds as alternative to control Staphylococus’s biofilms. However, this review is not well written, since it only has a superficial approach to the subject. In fact, it is not even mentioned the problem of antibiotic resistance, the division of the plant-derived compounds in groups, and, even only a little, the mechanism of action of those compounds. The manuscript only reports a table with some examples of plant-derived antimicrobial compounds, but it is not explained how each compound acts. Thus, it is a weak revision paper.

In addition, there are already several review articles in the literature focusing on this subject and, more importantly, in a much more complete way.

Specific comments:

Abstract:

The abstract begins to talk about Staphylococcus epidermidis and it seems that all the manuscript is about this strain. However, in the title and in keywords, authors also refer to Staphylococci or to Staphylococcus aureus.

Introduction:

This section addresses matters in a very superficial way. It is described the stages of biofilm formation, but, in my opinion, the more important subjects (abovementioned) are not even mentioned. For instance, the problem of antibiotics resistance, which is what leads to the need for the use of other alternatives as plant derivatives.

Author Response

Dear Reviewer we are more than happy to include your comments in order to improve our work.

Here’s our point-by-point answer.

The article by Svolana et al. and entitled “Sensitivity of staphylococcal biofilm to selected compounds of plants origin” is a review article that aims to describe a series of plant-derived compounds as alternative to control Staphylococus’s biofilms. However, this review is not well written, since it only has a superficial approach to the subject. In fact, it is not even mentioned the problem of antibiotic resistance, the division of the plant-derived compounds in groups, and, even only a little, the mechanism of action of those compounds. The manuscript only reports a table with some examples of plant-derived antimicrobial compounds, but it is not explained how each compound acts. Thus, it is a weak revision paper.

In addition, there are already several review articles in the literature focusing on this subject and, more importantly, in a much more complete way.

Authors’ reply: We reviewed the document with critical eye. Indeed, we found lot of imrovements and used them.

Abstract:

The abstract begins to talk about Staphylococcus epidermidis and it seems that all the manuscript is about this strain. However, in the title and in keywords, authors also refer to Staphylococci or to Staphylococcus aureus.

Authors’ reply:  We improved the abstract in order to reflect the work done.

Introduction:

This section addresses matters in a very superficial way. It is described the stages of biofilm formation, but, in my opinion, the more important subjects (abovementioned) are not even mentioned. For instance, the problem of antibiotics resistance, which is what leads to the need for the use of other alternatives as plant derivatives

Authors’ reply:  We tried to improve the Introduction. Moreover, we focused on biofilm formation. Also, added some comments in Conclusions.

Reviewer 2 Report

The subject of the manuscript is of current interest and importance. The manuscript needs a thorough revision by the authors. In the current format it shows a lack of work. The authors are supposed to review the action of plant components on staphylococcus biofilms. However, this section in the manuscript represents one paragraph and all the authors do is include a table. The authors need to develop this section in order for the article to be considered a review. In the same vein, the authors should modify the conclusion. It is really a kind of short discussion of results.

Other comments:

Line 15: A single?

Line 15: Change flora by microbiota

Line 15: The content of the review is not reflected in the abstract. At no point is there any mention of the alternatives to the antibiotics reviewed here. Please revise it.

Line 33-36: Please revise this sentence, some structure and grammatical changes are needed.

Line 39: ca. ?

Line 50: MRSA is just defined, don´t need to be defined between brackes.

Line 62: Microbiota.

Line 81: Microbiota.

Line 104: Please delete – and correct the sentence.

From line 154 to line 174 there is no references and there is a lot of information. Please add some references.

Line 174-189: The same comment. No references.

I recommend to the authors divide section 2. Staphylococcal biofilm in different subsections, maybe related with the different steps of biofilm formation. As it is structured now, the section is too long and difficult to read. Also, consider to include some kind of figure to have a graphical overview of Staphylococcus biofilm formation.

In table 1 the authors should include what Staphylococcus species were tested in the study and the number and source of strains. 

Author Response

Dear Reviewer we are more than happy to include your comments in order to improve our work.

Here’s the point-by-point answer.

The subject of the manuscript is of current interest and importance. The manuscript needs a thorough revision by the authors. In the current format it shows a lack of work. The authors are supposed to review the action of plant components on staphylococcus biofilms. However, this section in the manuscript represents one paragraph and all the authors do is include a table. The authors need to develop this section in order for the article to be considered a review. In the same vein, the authors should modify the conclusion. It is really a kind of short discussion of results.

Authors’ reply: We reviewed the document with critical eye. Indeed, we found lot of imrovements and used them.

Other comments:

Line 15: A single?

Authors’ reply: removed

Line 15: Change flora by microbiota

Authors’ reply: done

Line 15: The content of the review is not reflected in the abstract. At no point is there any mention of the alternatives to the antibiotics reviewed here. Please revise it.

Authors’ reply: Revised, added paragraph.

Line 33-36: Please revise this sentence, some structure and grammatical changes are needed.

Authors’ reply: Reworked.

Line 39: ca. ?

Authors’ reply: Removed.

Line 50: MRSA is just defined, don´t need to be defined between brackes.

Authors’ reply: Removed.

Line 62: Microbiota.

Authors’ reply: done

Line 81: Microbiota.

Authors’ reply: done

Line 104: Please delete – and correct the sentence.

Authors’ reply: done

From line 154 to line 174 there is no references and there is a lot of information. Please add some references.

Authors’ reply: done

Line 174-189: The same comment. No references.

Authors’ reply: done

I recommend to the authors divide section 2. Staphylococcal biofilm in different subsections, maybe related with the different steps of biofilm formation. As it is structured now, the section is too long and difficult to read. Also, consider to include some kind of figure to have a graphical overview of Staphylococcus biofilm formation.

Authors’ reply: We rewrote the paragrph and put some structure, ideed, now it is more clear to the reader. Good idea with the figure-  we implemented.

In table 1 the authors should include what Staphylococcus species were tested in the study and the number and source of strains. 

Authors’ reply: done

Round 2

Reviewer 1 Report

Although I continue to consider that this review article has some problems before, the authors have changed some parts and managed to improve it a little. Even so, and before its acceptance, I still have some comments.

Specific comments

Abstract

Please remove the sentence “It has been shown that bacterial cells in biofilm are more resistant to antimicrobial agents, including those of plant origin compared to cells growing in planktonic form” because this is already common sense. It was not demonstrated by this manuscript.

Introduction

I don’t understand why authors included more information, ie, detailed more, about biofilm formation. This review is not about that! I even said that the information about this subject was already too much. Please, resume that part.

Conclusions

I think that the text that the authors added to this section does not add anything relevant, so it is not necessary.

Author Response

Dear Reviewer,

Thank you very much for your effort to improve the manuscript.

We went through the manuscript and included your suggestions.

Here, you can find our answers.

Although I continue to consider that this review article has some problems before, the authors have changed some parts and managed to improve it a little. Even so, and before its acceptance, I still have some comments.

Specific comments

Abstract

Please remove the sentence “It has been shown that bacterial cells in biofilm are more resistant to antimicrobial agents, including those of plant origin compared to cells growing in planktonic form” because this is already common sense. It was not demonstrated by this manuscript.

 We removed it.

Introduction

I don’t understand why authors included more information, ie, detailed more, about biofilm formation. This review is not about that! I even said that the information about this subject was already too much. Please, resume that part.

Dear reviewer, we made the corrections you mentioned at the special request of the other reviewer, especially by adding the figure. However, we agree with your opinion and have made appropriate corrections and shortcuts to balance the two independent sometimes opposing remarks. Hopefully, it could satisfy your vision.

Conclusions

I think that the text that the authors added to this section does not add anything relevant, so it is not necessary.

We removed this.

Reviewer 2 Report

With the inclusion of a more elaborate part on the use of plant components in the treatment of biofilms, as well as the figure on biofilm formation, the manuscript has been greatly improved. The manuscript can be now accepted. 

Author Response

Dear Reviewer, Thank you very much for any comments that have allowed us to improve the manuscript.